# Munc13-1 and Munc18-1 together prevent NSF-dependent de-priming of synaptic vesicles

Enqi He[1], Keimpe Wierda[1,†], Rhode van Westen[2], Jurjen H. Broeke[2], Ruud F. Toonen[1], L. Niels Cornelisse[2] & Matthijs Verhage[1,2]

Synaptic transmission requires a stable pool of release-ready (primed) vesicles. Here we show that two molecules involved in SNARE-complex assembly, Munc13-1 and Munc18-1, together stabilize release-ready vesicles by preventing de-priming. Replacing neuronal Munc18-1 by a non-neuronal isoform Munc18-2 (Munc18-1/2SWAP) supports activity-dependent priming, but primed vesicles fall back into a non-releasable state (de-prime) within seconds. Munc13-1 deficiency produces a similar defect. Inhibitors of N-ethylmaleimide sensitive factor (NSF), N-ethylmaleimide (NEM) or interfering peptides, prevent de-priming in munc18-1/2SWAP or munc13-1 null synapses, but not in CAPS-1/2 null, another priming-deficient mutant. NEM rescues synaptic transmission in munc13-1 null and munc18-1/2SWAP synapses, in acute munc13-1 null slices and even partially in munc13-1/2 double null synapses. Together these data indicate that Munc13-1 and Munc18-1, but not CAPS-1/2, stabilize primed synaptic vesicles by preventing NSF-dependent de-priming.

[1] Department of Functional Genomics, Center for Neurogenomics and Cognitive Research, Neuroscience Campus Amsterdam, Vrije Universiteit (VU), Amsterdam 1081HV, The Netherlands. [2] Department of Clinical Genetics, Center for Neurogenomics and Cognitive Research, Neuroscience Campus Amsterdam, VU Medical Center, Amsterdam 1081HV, The Netherlands. † Present address: VIB Center for the Biology of Disease, Leuven 3000, Belgium; Center for Human Genetics, KU Leuven, 3000 Leuven, Belgium. Correspondence and requests for materials should be addressed to M.V. (email: matthijs@cncr.vu.nl).

In the mammalian central nervous system, synaptic transmission typically proceeds within milliseconds after incoming action potentials (APs). This property depends critically on the availability of a pool of synaptic vesicles that is immediately available for exocytosis, 'the readily releasable pool (RRP)' or the 'primed' vesicles. Despite intense research, it is currently not known how this primed vesicle pool is made and maintained.

The mammalian SNARE complexes that drive synaptic exocytosis consist of Syntaxin1, Synaptobrevin2 and SNAP25 (refs 1,2). During exocytosis, a four helical bundle is formed in a three step process, involving the N-terminal-, C-terminal and transmembrane areas[1,2]. One hypothesis proposes that primed vesicles have partially zippered *trans*-SNARE complexes (or 'SNAREpins') between vesicle and plasma membrane where only the N-terminal regions are assembled as a stable, partially assembled intermediate[3–5]. However, plausible alternatives for this model have been suggested[3,6]. Moreover, it is not clear which molecular processes stabilize this proposed partially assembled state, that is, prevent their instant full assembly or their complete disassembly.

Several molecular components have been identified that orchestrate SNARE-complex assembly, stability and disassembly. First, Munc13, Munc18 and CAPS are involved in assembly[7–10]. The number of primed vesicles and the priming rate are regulated by activity ($Ca^{2+}$) and modulatory signals like diacylglycerol and post-translational modifications[11]. Second, *N*-ethylmaleimide sensitive factor (NSF) and α/βSNAP disassemble post-fusion *cis*-complexes[12–14]. Finally, complexins and synaptotagmins have been proposed to prevent premature full assembly of partially assembled SNARE complexes[5,15,16]. However, the molecular factors that prevent disassembly of partially assembled SNARE complexes in primed vesicles (de-priming) are unknown and de-priming remains a poorly characterized process.

Here, we investigated the stability of the primed state and the priming/de-priming balance in mouse hippocampal synapses. We found that primed vesicles are instable in the absence of Munc13-1 or Munc18-1 and fall back into a non-releasable state (de-prime) within seconds. NEM or an NSF-inhibiting peptide prevented this de-priming in synapses that lack Munc13-1 or Munc18-1, but not in CAPS-1/2-deficient synapses. Hereby, preventing de-priming rescued synaptic transmission in *munc13-1 null*, in synapses where Munc18-1 had been replaced by Munc18-2 and even partially in *munc13-1/2 double null* synapses.

## Results

**Replacing Munc18-1 with -2 produces a *munc13-1 KO* phenocopy.** Previous data suggested that Munc18-1, in addition to its established role in docking/SNARE-complex assembly[17–19] is also required downstream in the secretion pathway[20–22]. We isolated this aspect of Munc18-1 function, by replacing the endogenous *munc18-1* gene in mouse hippocampal neurons with the non-neuronal isoforms Munc18-2 or Munc18-3, by Munc18-2 or -3 expression in *munc18-1 null* neurons ('*munc18-1/2*SWAP', '*munc18-1/3*SWAP'), and *munc18-1 null* neurons expressing Munc18-1 (*munc18-1/1*SWAP) as control. In contrast to *munc18-1 null* neurons[23], *munc18-1/2*SWAP and *munc18-1/3*SWAP neurons are viable and have a normal morphology and synapse density (Supplementary Fig. 1). However, *munc18-1/2*SWAP and *munc18-1/3*SWAP neurons showed severe defects in basal synaptic transmission: reduced evoked excitatory postsynaptic current (EPSC) amplitude (Fig. 1a,b) and miniature EPSC (mEPSC) frequency, but not-amplitude (Fig. 1c-e). Vesicle docking was largely rescued by Munc18-2, but poorly by Munc18-3 (Fig. 1f-k). Hence, replacing Munc18-1

by Munc18-2 or -3 fully supports neuronal viability but not synaptic transmission.

While basal transmission was severely reduced, 100 APs at 40 Hz potentiated the evoked EPSC amplitude by 500% in *munc18-1/2*SWAP synapses, but not in *munc18-1/3*SWAP synapses (Fig. 1l). This strong potentiation in *munc18-1/2*SWAP synapses was similar to potentiation previously observed in *munc13-1 null* glutamatergic synapses[7] and *CAPS-1/2 null* synapses[10]. Applying hypertonic sucrose solution to probe the RRP size in *munc18-1/2*SWAP synapses revealed a sevenfold smaller RRP as compared to control (*munc18-1/1*SWAP) synapses. A second application of hypertonic sucrose, 3 s after the first, detected virtually no RRP in contrast to control[24] (Fig. 1m). However, when 100AP at 40 Hz stimulation was applied between the two hypertonic sucrose applications, the RRP detected by the second application was 37% increased in *munc18-1/2*SWAP neurons (Fig. 1m,n). This suggests that a stable RRP, but not priming *per se*, is abolished in *munc18-1/2*SWAP synapses, as previously concluded for *munc13-1 null* and *CAPS-1/2 null* synapses[7,10]. Hence, replacing Munc18-1 by Munc18-2 produces a phenocopy of *munc13-1 null* and *CAPS-1/2 null* synaptic phenotype and confirms a post-docking role of Munc18-1 in synaptic transmission that cannot be compensated for by Munc18-2.

**Primed vesicles rapidly de-prime without Munc13-1/Munc18-1.** The activity-induced potentiation of synaptic transmission in *munc18-1/2*SWAP synapses was transient, as previously observed for *munc13-1 null* and *CAPS-1/2 null* synapses[7,10]. We investigated this transient nature of fusion-competent (primed) vesicles in *munc18-1/2*SWAP, *munc13-1 null* and *CAPS-1/2 null* synapses by single test stimuli at different delays after activity-induced potentiation ('priming train', 100 APs at 40 Hz). The largest potentiation of evoked EPSCs was observed 3 s after priming trains and then decayed (Fig. 2a,b). The time course of this decay was strikingly similar among all three genotypes (Fig. 2b).

To quantify the number of fusion-competent vesicles that disappeared during this decay, we compared the total number of vesicles released by 10 stimuli at different intervals after priming trains in *munc18-1/2*SWAP and *munc13-1 null* synapses. We subtracted the total charge induced by 10 stimuli 45 s after priming trains from the total change at 10 s after priming trains (Fig. 2c, see Experimental procedures for details). In addition to this 10–45 s interval, we also analysed the 3–30 s interval (Supplementary Fig. 2a–e). These comparisons revealed that very few vesicles that were fusion-competent shortly after priming trains (3 or 10 s after the train), were still fusion-competent after 30 or 45 s (Fig. 2d,e and Supplementary Fig. 2a–e). This loss of fusion-competent vesicles cannot be explained by spontaneous release. Spontaneous release is known to be increased after 100 APs at 40 Hz, but the total spontaneous release during the 35 s interval between the 10 and 45 s time points was only 27% of the synchronous (evoked) release lost between these time points (Fig. 2f,g). We also tested these protocols in wild-type (WT) neurons and neurons over-expressing NSF. No evidence for de-priming was observed in these cases (Supplementary Fig. 3a–d). This suggests that the majority of the vesicles primed by AP-trains, de-primed in in *munc18-1/2*SWAP and *munc13-1 null* synapses, but not in WT neurons or neurons over-expressing NSF.

We also assessed the total pool of releasable vesicles with a $Ca^{2+}$-independent method, using hypertonic sucrose to exclude contributions of $Ca^{2+}$-dependent priming during AP-trains on our estimate of fusion-competent vesicles (Supplementary

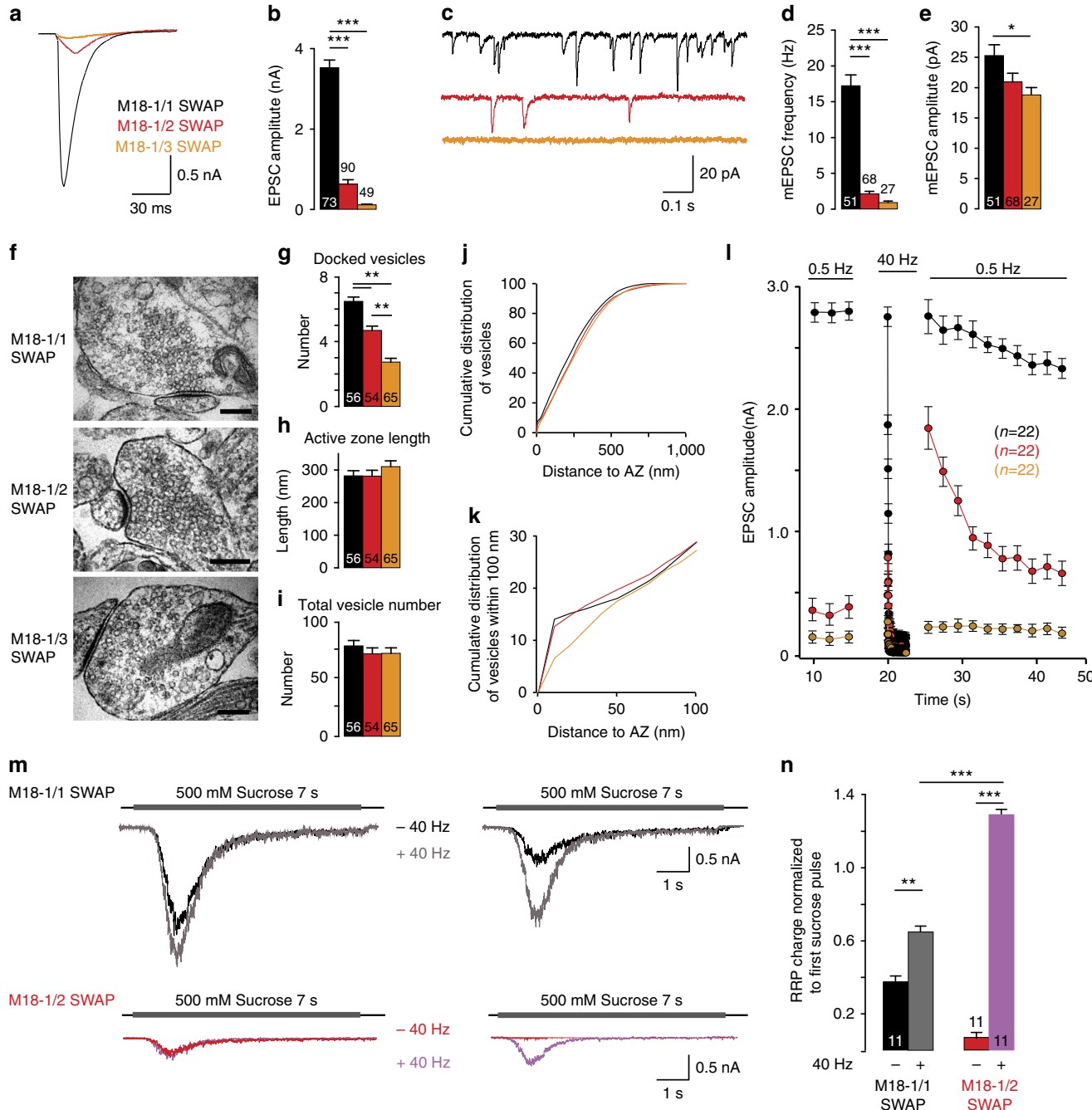

**Figure 1 | Replacing Munc18-1 with Munc18-2 but not Munc18-3 produces a *munc13-1 null* phenocopy.** (**a,b**) Sample traces and quantification of evoked EPSCs in naive *munc18-1/1SWAP*, *munc18-1/2SWAP* and *munc18-1/3SWAP* neurons. (**c–e**) Mini EPSCs sample traces (**c**), and quantification of frequency (**d**) and amplitudes (**e**). (**f–k**) Ultrastructural analysis of *munc18-1/1SWAP*, *munc18-1/2SWAP* and *munc18-1/3SWAP* asymmetric (glutamatergic) synapses: sample electron micrographs; scale bar,100 nm, (**f**) and quantification of docked vesicles (**g**), active zone length (**h**) and total vesicle number (**i**); cumulative plot of vesicle distance from the active zone (**j**) and a zoom of the first 100 nm (**k**). (**l**) Synaptic run down during 100AP at 40 Hz and transient potentiation of synaptic transmission in *munc18-1/2SWAP*, but not the other genotypes. (**m,n**) Assessment of activity-dependent priming using application of hypertonic sucrose to probe the readily releasable vesicle pool: sample traces (**m**) of *munc18-1/2SWAP* (red/blue) and control (*munc18-1/1SWAP*, black/grey) neurons stimulated with 500 mM sucrose twice with a 3 s interval with (grey/bleu) or without (black/red) 100AP at 40 Hz during this interval; (**n**) quantification of total charge generated by second hypertonic sucrose pulses, normalized to the first pulse. All data in this figure are means ± s.e.m.; *$P < 0.05$, **$P < 0.01$, ***$P < 0.001$ as determined by ANOVA. See Supplementary Table 1 for all values, s.e.m. and *n*-numbers plotted in this figure. ANOVA, analysis of variance.

Fig. 2f). Thirty seconds after priming trains, the reduction in sucrose-evoked responses was comparable with the reduction in AP-evoked responses (compare Supplementary Fig. 2g with 2c), and much larger than the loss of vesicles through spontaneous release (Supplementary Fig. 2d,e,h,i). Hence, fusion-competent

vesicles that contributed to synaptic transmission shortly after priming trains, no longer did so after 30 or 45 s in the absence of Munc13-1 or Munc18-1. This confirms that the majority of the vesicles primed by AP-trains, de-primed in the absence of these molecules.

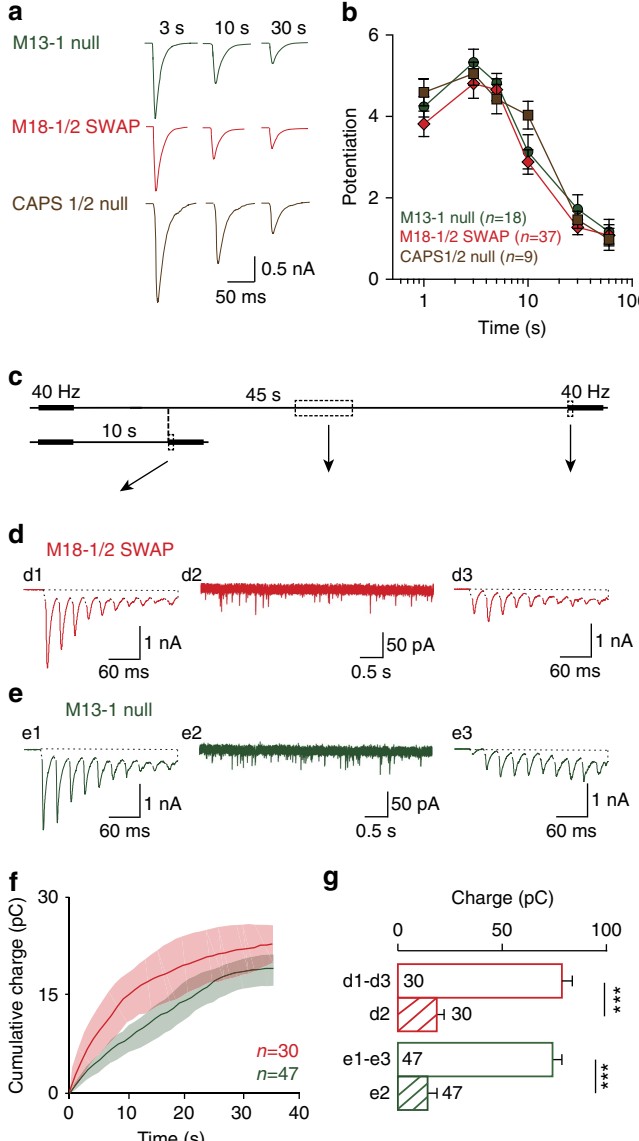

**Figure 2 | Primed vesicles rapidly de-prime in the absence of Munc13-1 or Munc18-1.** (**a**) Sample traces of evoked EPSCs following 3, 10 and 30 s after a priming train (100AP at 40 Hz). (**b**) Similar time course of activity-dependent potentiation in *munc18-1/2SWAP*, *munc13-1 null* and *CAPS-1/2 null* synapses. (**c**) Paradigm to assess de-priming (see Methods for details). (**d,e**) sample traces of dual 100AP at 40 Hz with 10 or 45 s interval in *munc18-1/2SWAP* (**d**) and *munc13-1 null* (**e**) synapses. (**f**) Spontaneous fusion events were quantified for the 10–45 s intervals. (**g**) To quantify de-priming, the total charge of the first 10 responses (d1, d3; e1, e3) of each stimulus train was quantified. The spontaneous fusion of vesicles in the 10–45 s intervals (d2; e2) cannot explain the loss of fusion-competent vesicles after 45 s, defined as the difference in total charge between 10 and 45 s intervals (d1–d3; e1–e3). All data in this figure are means ± s.e.m., ***$P < 0.001$, P value are determined by Wilcoxon signed rank test; the de-priming was also observed with 3–30 s intervals (Supplementary Fig. 2).

## NSF-inhibition rescues de-priming defects.
One current working model proposes that in the primed state, a *trans*-SNARE-complex is already established 'loosely'[25] or 'partially'[26,27]. Therefore, we hypothesized that de-priming observed in the *munc18-1/2SWAP*, *munc13-1* and *CAPS-1/2 null* synapses involves *trans*-SNARE-complex disassembly. To test this, we applied *N*-ethylmaleimide (NEM), a generic NSF inhibitor, and specific NSF-interfering peptides in *munc13-1 null*, *munc18-1/2SWAP* and *CAPS-1/2 null* synapses.

Within 10 s after NEM application, the small initial EPSC of *munc13-1 null* synapses was potentiated nearly 500%, to a similar extent as high-frequency stimulation (Fig. 3a,b), and the potentiation is time- and dose-dependent (Supplementary Fig. 4c,d). The mEPSC frequency also increased, by 250%, with no significant change in mEPSC amplitude (Fig. 3c,d). In *munc18-1/2SWAP* synapses, a similar potentiation of evoked and spontaneous release was observed (Fig. 3e,f). Strikingly, NEM did not potentiate synaptic transmission in *CAPS-1/2 null* synapses (Fig. 3g,h). This lack of effect in *CAPS-1/2 null* synapses confirms that NEM does not rescue priming-deficient synapses by elevating the cellular calcium concentration (as 40 Hz stimulation does). Furthermore, NEM did not potentiate synaptic transmission in WT neurons (Supplementary Fig. 4a,b), ruling out nonspecific enhancement of transmission, unrelated to de-priming.

The effect of NEM on synaptic transmission in *munc13-1 null* synapses built up during the first seconds after NEM application, was highest 10–60 s after application, lasted far beyond activity-induced potentiation and then decayed (Fig. 3i and Supplementary Fig. 4c–f). This final decay can be explained by inhibition of NSF's established role in *cis*-SNARE disassembly and recycling (and/or to reactivate release sites). Taken together, these data show that NSF-inhibition prevents de-priming in *munc18-1/2SWAP* and *munc13-1 null* synapses, but not in *CAPS-1/2 null* synapses, and restored their ability to generate near-normal EPSCs upon stimulation.

In contrast to *munc13-1 null* synapses, *munc13-1/2 double null* synapses lack all synaptic activity[28]. Remarkably, after NEM application to *munc13-1/2 null* synapses single test stimuli now produced a small EPSC (Fig. 3j,k). NEM did not produce similar effects in mutants known to produce more upstream defects in the secretory pathway, *munc18-1* conditional *null* and syntaxin-deficient (BoNT-C treated) synapses (Fig. 3l,m). Hence, synapses deficient for proteins that act upstream of priming (that is, tethering/docking) are not rescued by NSF-inhibition, but NSF-inhibition in *munc13-1/2 double null* synapses reveals synaptic vesicles prime in the absence of Munc13-1 and Munc13-2.

To obtain independent proof for the role of NSF in de-priming, we tested specific NSF (peptide) blockers[13,29]. Peptide loading via the patch pipette for 30 min or application of cell membrane permeable peptide (TAT-NSF peptide[30], potentiated EPSCs in *munc13-1 null* synapses, similar to NEM (Fig. 4a–d). The enhancement of synaptic transmission was more variable than NEM application, probably due to the slow and incomplete loading of the peptide into synapses. Peptide loading, via the pipette or using the TAT-tag, is inherently slower than NEM application. On this slower timescale it is not possible to completely separate de-priming inhibition from inhibition of NSF's established role in *cis*-SNARE disassembly and recycling (and/or to reactivate release sites). However, both approaches yielded robust significant enhancement of synaptic transmission in mutant synapses confirming that NSF-inhibition prevents de-priming.

## N-ethylmaleimide enhances activity in *munc13-1 null* networks.
To confirm that de-priming in the absence of Munc13-1 also occurs in intact neural networks, we tested NEM in acute hippocampal slices of newborn *munc13-1 null* mice. At birth, the hippocampal synaptic network is still developing, many synapses are electro-physiologically silent and little evoked activity can be recorded[31,32]. However, spontaneous activity can be detected, also

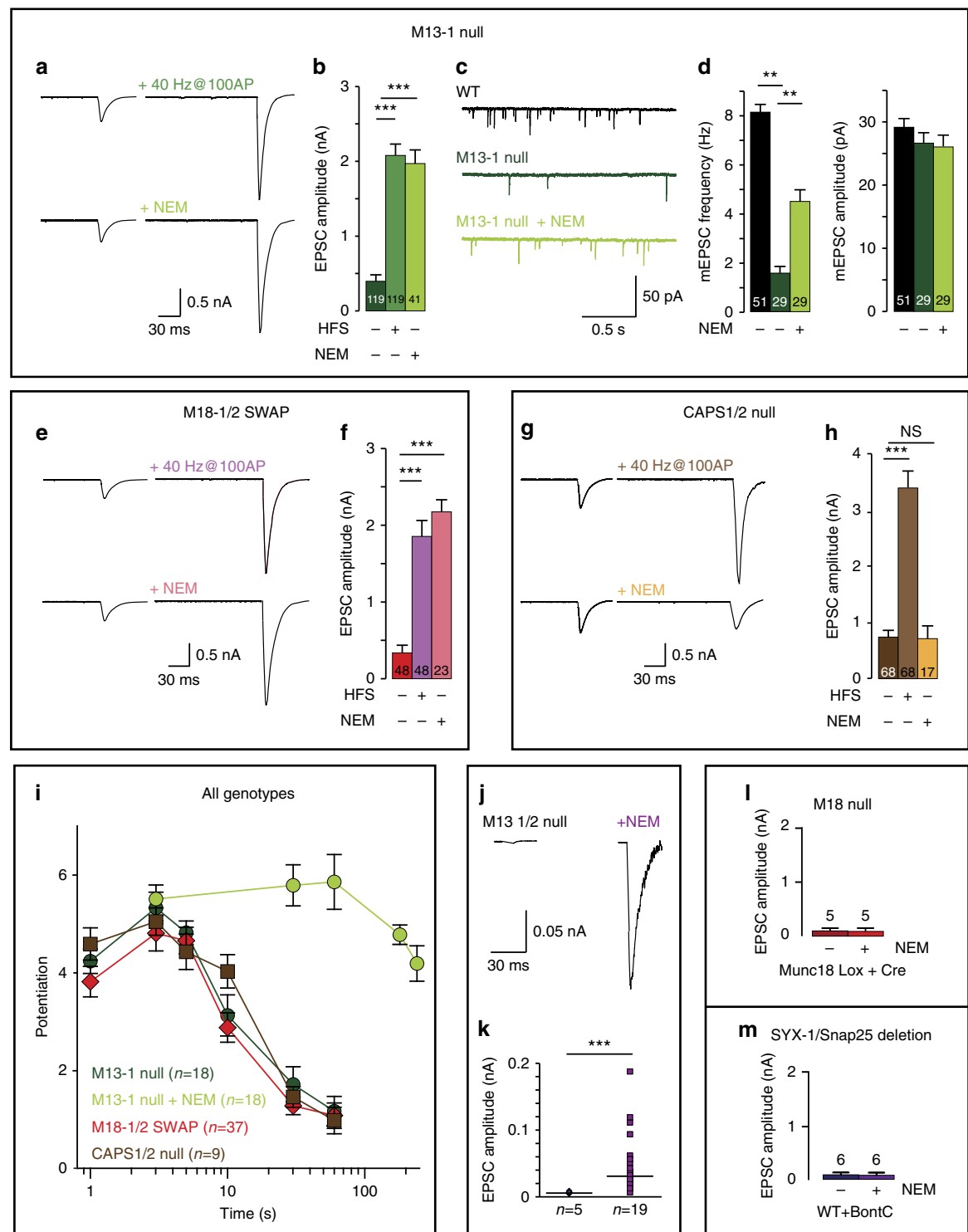

**Figure 3 | NSF-inhibition prevents de-priming and restores synaptic transmission in *munc18-1/2SWAP* and *munc13-1 null*.** (**a,b**) Sample traces and quantification of EPSCs potentiated by 100AP at 40 Hz or 10 s NEM application (100 μM) in *munc13-1 null* synapses. (**c,d**) mEPSCs sample traces and quantification of frequency and amplitude. (**e,f**) Sample traces and quantification of EPSCs potentiated by 100AP at 40 Hz or 10 s NEM application (100 μM) in *munc18-1/2SWAP* synapses. (**g,h**) Sample traces and quantification of EPSCs potentiated by 100AP at 40 Hz or 10 s NEM application (100 μM) in *CAPS-1/2 null* synapses. (**i**) Comparison of effectiveness between 100AP at 40 Hz or 10 s NEM application on potentiation of synaptic transmission in all genotypes. Data on 100AP at 40 Hz stimulation are copied from Fig. 2a for comparison. (**j,k**) Sample traces and quantification of EPSCs in *munc13-1/2 null* neurons with/without 10 s NEM application; (**l**) quantification of EPSCs in Cre-expressing *munc18-1 lox* neurons with and without NEM application. (**m**) Quantification of EPSCs in BoNT-C infected WT neurons with and without NEM application. All data in this figure are means ± s.e.m. In **b,d,f,h**, *$P < 0.05$, **$P < 0.01$, ***$P < 0.001$, are determined by ANOVA; in **k**, *P* values are determined by a binomial test. See Supplementary Table 1 for all values, s.e.m. and *n*-numbers plotted in this figure. ANOVA, analysis of variance.

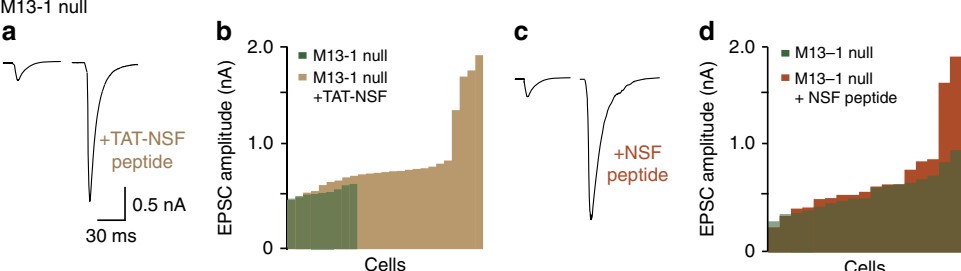

**Figure 4 | NSF-specific blockers restore synaptic transmission in *munc13-1 null* synapses.** (**a**,**b**) Sample traces and histogram of EPSCs in *munc13-1 null* neurons with and without application of TAT-tagged NSF inhibitor peptide. (**c**,**d**) Sample traces and histogram of EPSCs in *munc13-1 null* neurons with or without application of NSF-inhibiting peptide through the pipette. *P* values are determined by binomial test, \*\*\**P* < 0.001.

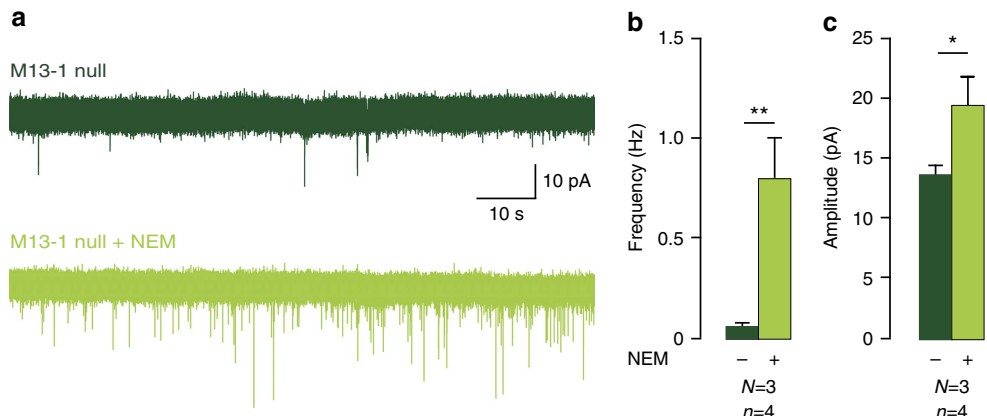

**Figure 5 | Potentiation of synaptic responses by NSF-inhibition in acute brain slices.** (**a**) Sample traces of spontaneous activity in immature synapses of newborn hippocampal slices of *munc13-1 null* mice with/without NEM (100 μM). (**b**,**c**) Quantification of mEPSC frequency and amplitude, $N = 3$, $n = 4$. Data in this figure are means ± s.e.m. \**P* < 0.05, \*\**P* < 0.01, *P* value are determined by paired *t*-test.

in munc13-1 null slices (Fig. 5a). NEM application increased spontaneous activity by more than 700% (Fig. 5a,b), that is, a larger increase than observed in cultured neurons (<300%, compare Figs 3d and 5b). NEM also produced a moderate increase in the amplitude of spontaneous events in these developing networks (Fig. 5c). These data confirm that NSF-inhibition prevents de-priming in *munc13-1 null* synapses also in acute *ex vivo* preparations.

## Discussion

This study shows that Munc13-1 and Munc18-1, in addition to their established role in docking/priming vesicles for fusion, prevent de-priming. Without this aspect of Munc13/18 function, synapses are very weak and activity can potentiate synaptic transmission only transiently. This suggests that de-priming is an important aspect of the synaptic vesicle cycle. Preventing de-priming is a crucial aspect of synaptic transmission.

The established (forward) role of Munc13-1 and Munc18-1 in docking/priming is shared with other isoforms, Munc13-2 and Munc18-2, and these isoforms rescue docking/priming in the absence of Munc13-1 and Munc18-1. However, the ability to inhibit de-priming is specific for the synaptic isoforms, Munc13-1 and Munc18-1, suggesting this function is an evolutionary adaptation that helps synapses to build stable reservoirs of fusion-ready vesicles to sustain high-frequency transmission. Our data suggest that Munc13-1 and Munc18-1 are both required to prevent de-priming (Fig. 6a), probably by acting in tandem.

This is consistent with their concerted actions in synaptic plasticity[21,33]. The fact that synaptic transmission is largely rescued by acute NSF-inhibition in *munc13-1 null* synapses suggests that de-priming inhibition is a major aspect of Munc13-1 function.

Our findings are in line with kinetic schemes considering the RRP as a dynamic equilibrium between priming and de-priming rates[34,35]. The RRP is generally defined as the pool of vesicles having a release probability $> 0$. However, substantial heterogeneity exists among vesicles in the RRP and activity and/or modulatory signals change the likeliness that a given vesicle fuses upon a given $Ca^{2+}$-increase (fusogenicity), probably due to modulation of the energy barrier for fusion[36,37]. It is already well documented that a $Ca^{2+}$-dependent increase in the forward priming rate and presynaptic modulators such as diacylglycerol promote fusogenicity (and in some cases also the RRP size), and induce synaptic plasticity[21,33,36,38]. Plausible mechanisms for this modulation are an increase in the number of SNARE-complexes (partially) assembled per vesicle[37,39] or an increased efficiency of the fusion machinery, such as Munc13s (refs 38,40,41). In addition to the regulation of the forward priming rate, we now present evidence that de-priming is also regulated by presynaptic proteins. We show that regulation of de-priming also regulates fusogenicity and RRP size in an equilibrium with forward priming rates. This idea is in line with observations in chromaffin cells, where the RRP scales with basal calcium and transiently facilitates (overshoots) after stimulation[42]. This suggests a high de-priming rate, causing a

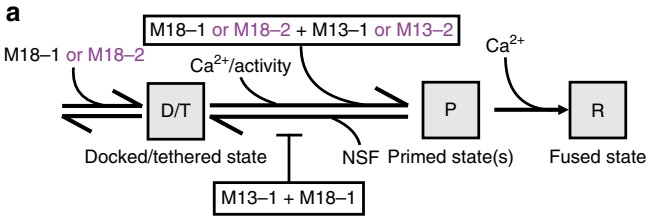

**a**

**b**

|  | Cell viability | Tethering/ docking | Priming | Prevent de-priming |
|---|---|---|---|---|
| Munc18–1 | +++ | +++ | +++ | +++ |
| Munc18–2 | +++ | ++ | +++ | − |
| Munc18–3 | +++ | + | − | − |

**Figure 6 | Working model for how Munc13-1 and Munc18-1 prevent NSF-dependent de-priming and comparison of the function of Munc13/18 isoforms in the secretory pathway** (**a**) Flow chart working model of the proposed actions of Munc13, Munc18 in the secretory pathway. Dominant isoforms are depicted in black and non-dominant/heterologous isoforms in purple. (**b**) Summary of proposed (non-) redundancy of cellular functions among Munc18 isoforms 1, 2 and 3 on cell viability, vesicle tethering/docking, priming, prevent de-priming. Number of + indicates to what extent a given isoform supports a given function, where +++ is full support and − indicates no support.

relatively low RRP size under resting, low $Ca^{2+}$ conditions, which shifts (temporarily) to a larger RRP when calcium levels are higher. Furthermore, a similar mechanism was suggested in general models for short-term plasticity observed in crayfish neuromuscular junctions and hippocampal- and cerebellar mossy fibre boutons[35,43]. Strikingly, in this model[35], the de-priming rate is the only parameter that is different between depressing 'phasic' synapses and facilitating 'tonic' synapses[35,44]. This theoretical prediction fits very well with the switch from depression to facilitation demonstrated here by increasing de-priming in M18-1/2SWAP and Munc13-1 KO neurons and also with the fact that NEM application produces a (mild) potentiation of synaptic transmission in mammalian neuromuscular junctions[45–47]. In these synapses, vesicles still fuse in the absence of Munc13-1/2, but there is no stable RRP[48]. Hence, expression of specific Munc13- (and possibly also Munc18-) isoforms probably defines de-priming rates in synapses and hereby their depressing or facilitating properties. Indeed, differential expression of Munc13-2 in basolateral amygdala synapses is responsible for a shift in facilitation versus depression[49]. However, dentate gyrus mossy fibre synapses that prominently express Munc13-2, but in this case also Munc13-1, become even more facilitating upon deletion of Munc13-2 expression[50]. This remains a puzzling observation, especially why Munc13-1 expression does not lead to a higher initial release probability in these synapses. The ability of Munc13-1 to inhibit de-priming may depend on other molecules, such as RIM, RIM-BPs and ELKS. Deletion of RIM and RIM-BP switches depressing synapses to facilitation, expression of a RIM mutant that specifically lacks Munc13-1 binding/activation shows facilitation[51–54] and ELKS1 specifically localizes Munc13-2 to specific active zones and determines facilitation/depression properties[55].

To obtain experimental evidence for the role of NSF in de-priming, required acute inhibition of NSF. During chronic inhibition, for example, shRNA knockdown, the canonical function of NSF in regenerating *cis*-SNARE complexes

throughout the cell prevails and neurons die within a few days (Supplementary Fig. 5a). The main evidence for a role of NSF in de-priming in this study comes from the use of NEM, an alkene reactive towards thiols that modifies cysteines in cellular proteins with little selectivity. Therefore, NEM may have nonspecific effects in addition to its established inhibition of NSF. However, at least five arguments suggest that such potential nonspecific effects do not confound our conclusions on the role of NSF in de-priming. First, NEM does not rescue synaptic transmission in *CAPS-1/2 null* synapses, which are equally weak as *munc13-1 null* synapses[10] and can, like *munc13-1 null* and *munc18-1/2SWAP* synapses, also be rescued by trains of APs. Hence, NEM does not simply promote activity or increase intracellular $Ca^{2+}$ to rescue *munc13-1 null* and *munc18-1/2SWAP* synapses. Second, the rescue effect of NEM lasts much longer than activity/$Ca^{2+}$-increases (Fig. 3i and Supplementary Fig. 4e–h). Third, NEM does not promote synaptic transmission in WT synapses (Supplementary Fig. 4a), Fourth, NEM does not rescue synapses with more upstream defects, that is, BoNT-C treated or *munc18-1 null* synapses (Fig. 3l,m). Finally, NSF-inhibition using independent approaches, using a specific inhibitory peptide, also rescues *munc13-1 null* and *munc18-1/2SWAP* synapses. Together, these findings confirm the specificity of this generic inhibitor.

The replacement experiments (*munc18-1/2SWAP*) help to define Munc18-1's post-docking role. The previously described role of Munc18-1 in the initial vesicle docking/tethering, upstream of SNARE-complex formation[17,19], can largely be covered by Munc18-2 expression as proposed before[20]. The fact that the *munc18-1/2SWAP* produces a *munc13-1* phenocopy now suggests that Munc18 acts in tandem with Munc13 to prime vesicles. Together these findings suggest that the elusive positive function of Munc18 in the vesicle cycle consist of (i) promoting docking/tethering, (ii) promoting priming in tandem with Munc13s and (iii) preventing de-priming (Fig. 6a). The first two aspects of Munc18-1 function can be covered by Munc18-2, the latter cannot. Expression of the more distant isoform, Munc18-3, in *munc18-1 null* neurons rescued cell viability, but not priming (in contrast to Munc18-2). This suggests that the role of Munc18-1 in cell viability is yet another distinct aspect of this pivotal molecule and that the three isoforms support these different aspects to a different extent (Fig. 6b).

Whether αSNAP/NSF act on *trans*-SNARE complexes has been subject of long debate[56]. Our study provides two lines of evidence, using the generic blocker NEM and specific interfering peptides that they indeed do. The fact that Munc13-1 and Munc18-1 prevent NSF from de-priming primed vesicles may explain why neuronal *trans*-SNARE-complexes *in situ* were found to be NSF-resistant.

## Methods

**Laboratory animals.** *Munc18-1 null* and *CAPS-1/2 null* mice were generated as described before[10,23] and maintained on a C57BL/6J background after >20 (CAPS-1/2) or >40 (munc18-1) generations of backcrossing. *Munc13-1 null,* and *Munc13-1/2 null* mice were produced as described previously[28,57] and maintained on a FVB/NHan background. All neurons were obtained from E18 embryos of all sexes by caesarean sections of pregnant females from timed heterozygous matings. Animals were housed and bred according to institutional and Dutch governmental guidelines, and all procedures are approved by the ethical committee of the Vrije Universiteit, Amsterdam, The Netherlands.

**Neuronal cultures and lentiviral transduction.** Hippocampi and cortices were separately collected in ice-cold Hanks Buffered Salt Solution (HBSS; Sigma) buffered with 1 mM HEPS (Invitrogen). After removal of the meninges, tissue was incubated in Hanks-HEPES with 0.25% trypsin (Invitrogen) for 12 min at 37 °C. After washing, the tissues were triturated and counted in a Fuchs-Rosenthal chamber. Neurons were plated in pre-warmed Neurobasal medium (Invitrogen) supplemented with 2% B-27 (Invitrogen), 1.8% HEPES, 0.25% glutamax

(Invitrogen) and 0.1% Pen/Strep (Invitrogen). To achieve autaptic cultures, hippocampal neurons were plated at a density of 6 K per well of a 12-well plate on micro-islands of rat glia on ultraviolet-sterilized agarose-coated etched glass coverslips stamped with a 0.1 mg ml$^{-1}$ poly-D-lysine (Sigma) and 0.2 mg ml$^{-1}$ rat tail collagen (BD biosciences) solution[58]. Network cultures were generated by plating cortical neurons(10–25 K per well on a 12-well plate) on a confluent layer of rat glia grown on etched glass coverslips sprayed with a 0.1 mg ml$^{-1}$ poly-D-lysine (Sigma) and 0.2 mg ml$^{-1}$ rat tail collagen (BD biosciences) solution. Neurons were infected at DIV0 with lentiviral particles encoding Munc18-1, Munc18-2 and Munc18-3 (ref. 21), and were allowed to develop for 13-18 days before measuring. For NSF knockdown experiments, the shRNA targeting mouse NSF was obtained from Amsterdam Medical Center (TRCN0000101667, Sigma Aldrich) and cloned into the same lentiviral backbone as the Munc18 cDNAs (see above). A knockdown-resistant NSF variant was developed by introducing silent mutations in the mouse NSF cDNA (5′-ccc ggg cgc ttg gaa gtt aaa-3′) and cloned into lentiviral backbone.

**Electron microscopy.** Autaptic hippocampal cultures obtained from three different litters were fixed at DIV14-16 for 45 min at room temperature with 2.5% glutaraldehyde in 0.1 M cacodylate buffer (pH 7.4)[19,21]. Only glia islands containing a single neuron were used for analysis. After fixation cells were washed three times for 5 min with 0.1 M cacodylate buffer (pH 7.4), post-fixed for 2 h at room temperature with 1% osmium tetroxide/1% potassium ferrocyanide in double distilled water, washed and stained with 1% uranyl acetate for 40 min in the dark. Following dehydration through a series of increasing ethanol concentrations, cells were embedded in Epon and polymerized for 24 h at 60 °C. After polymerization of the Epon, the coverslip was removed by alternately dipping it in liquid nitrogen and hot water. Cells of interest were selected by observing the flat Epon-embedded cell monolayer under the light microscope, and mounted on pre-polymerized Epon blocks for thin sectioning. Ultrathin sections (∼80 nm) were cut parallel to the cell monolayer and collected on single-slot, formvar-coated copper grids, and stained in uranyl acetate and lead citrate. Autaptic synapses were selected at low magnification using a JEOL 1010 electron microscope. All analyses were performed on single ultrathin sections of randomly selected synapses. The distribution of synaptic vesicles, total synaptic vesicle number and active zone length were measured with Image J (National Institute of Health, USA) on digital images of synapses taken at ×80,000 magnification using analySIS software (Soft Imaging System, GmbH, Germany). The observer was blinded for the genotype. For all morphological analyses we selected only synapses with intact synaptic plasma membranes with a recognizable pre- and postsynaptic density and clear synaptic vesicle membranes. Docked synaptic vesicles had a distance of 0 nm from the synaptic vesicle membrane to the active zone membrane. The active zone membrane was recognized as a specialized part of the presynaptic plasma membrane that contained a clear presynaptic density.

**Electrophysiology of autaptic neurons.** Whole-cell recordings were performed using an Axopatch 200B amplifier (Molecular Devices) at room temperature. Digidata 1440 and Clampex 10.0 (Molecular Devices) were used for data acquisition. The external solution contained the following (in mM): 140 NaCl, 2.4 KCl, 4 MgCl$_2$ 4 CaCl$_2$, 10 HEPES and 10 glucose (pH = 7.30, 300 mOsmol). Patch pipette solution contained the following (in mM): 125 K$^+$–gluconic acid, 10 NaCl, 4.6 MgCl$_2$, 15 creatine phosphate, 10 U ml$^{-1}$ phosphocreatine kinase and 1 EGTA (pH 7.30). Only cells with an access resistance of <10 MΩ and leak current of <300 pA were accepted for analysis. The recorded was compensated to 60%. The RRP size was determined by a rapid switch to external saline solution made hypertonic by the addition of 0.5 M sucrose for 7 s (ref. 37). All analyses were performed using Clampfit 10.2, MiniAnalysis (Synaptosoft) and custom-written software routines in Matlab R2010a. In AP-induced EPSC's, the stimulation artefact was removed and interpolated using cubic interpolation. The baseline for synchronous charge is determined by a straight line between the starting point of response $r$ and the starting point of response $r+1$. NEM (Sigma) was added to external solution at various concentrations (50, 100, 200, 500, 1,000 μM). NSF blocking peptide (Tebu-Bio) was loaded intracellularly via the patch pipette. The tat-NSF peptides were added to the cell media and incubated with neurons in the incubator for 15 min.

To measure the stability of activity-dependent primed vesicles, the number of primed vesicles at different delays after activity-induced potentiation was compared. To achieve this, evoked EPSCs were measured after test trains (100AP at 40 Hz) with variable delay after activity-dependent priming by a priming train (also 100AP at 40 Hz). We compared two intervals: (a) 3 versus 30 s after the priming train and (b) 10 versus 45 s after the priming train. The assessment of responses 3 s after the priming train was complicated by remaining asynchronous release resulting from the preceding priming train. Therefore the comparison 10–45 s was used in the main text and the comparison 3–30 s is shown as Supplementary Data. Potentiation of evoked release after the priming train was restricted to the first 10 EPSC, presumably because these are dominated by synchronous release that draws directly from the established RRP after the priming train, whereas asynchronous release also includes newly primed vesicles which becomes dominant in later responses. Therefore, we included only the charge of the first 10 EPSCs when comparing the pool of primed vesicles (expressed in nC) at

different intervals after the priming train (10 versus 45 and 3 versus 30). Furthermore, to quantify the number of vesicles that fused spontaneously during the test intervals, the spontaneous release charge was quantified between the two time points (10 versus 45 and 3 versus 30). The charge of each event was assessed by subtracting the baseline from the trace. The baseline was determined every 100ms as the midline of the noise. To avoid any confounding effect of activity-dependent priming on RRP estimation we performed an independent set of experiments where we used hypertonic sucrose (500 mM, see above) to measure the pool of primed vesicles 3 and 30 s after a priming train. De-priming was quantified by integrating the current obtained by subtracting the response at 30 s from the response at 3 s after the priming train.

**Preparation of acute slices.** Acute coronal slices were obtained from newborn (P0) mice. Brains were rapidly removed after decapitation and placed into ice-cold choline solution (110 mM choline chloride, 11.6 mM Na-ascorbate, 7 mM MgCl$_2$, 3.1 mM Na-pyruvate, 2.5 mM KCl, 1.3 mM NaH$_2$PO$_4$, 0.5 mM CaCl$_2$, 26 mM NaHCO$_3$, 10 mM glucose, ∼300 mOsm, pH 7.4) that was continuously perfused with carbogen (95% O$_2$ and 5% CO$_2$). Coronal slices of 350 μm thickness were retrieved using a using a vibrating-blade microtome (HM-650 V, Thermo-Scientific). Retrieved slices were placed in a holding chamber containing aCSF (125 mM NaCl; 3 mM KCl; 1.2 mM NaH$_2$PO4; 1 mM MgSO4; 2 mM CaCl$_2$; 26 mM NaHCO$_3$; 10 mM glucose, 300 mOsm, pH 7.4), oxygenated with 95% O$_2$, 5% CO$_2$. Slices were allowed to recover for 1 h following slicing.

**Electrophysiology of acute slices.** Whole-cell recordings were performed using an MultiClamp 700B amplifier (Molecular Devices) in a submerged holding chamber at room temperature under continuous superfusion of oxygenated (95% O$_2$ and 5% CO$_2$) aCSF. Digidata 1440A and Clampex 10.2 (Molecular Devices) were used for data acquisition. Recordings were sampled at 10 kHz and low-pass filtered at 3 kHz. Borosilicate patch pipettes (resistance 2.5–4.5 MΩ) were filled with intracellular solution (148 mM K-gluconate, 1 mM KCl, 10 mM HEPES, 4 mM Mg-ATP, 4 mM K$_2$-Phosphocreatine, 0.4 mM GTP, pH 7.4, ∼290mOsm). Neurons were held in voltage-clamp at −70 mV. Only cells with an access resistance of <20 MΩ and leak current of <100 pA were accepted for analysis. Spontaneous events were analysed in MiniAnalysis software 6.0 (Synaptosoft, NJ, USA).

**Immunocytochemistry.** Cultures were fixed with 3.7% formaldehyde (Electron Microscopy Sciences). After washing with PBS, cells were permeated with 0.1% Triton X-100 for 5 min and incubated in 2% normal goat serum for 20 min to block aspecific binding. Cells were incubated for 1 h at room temperature in a mixture of monoclonal mouse anti-VAMP (1:1,000, SySy, 104204) or ant-syntaxin (1:1,000, SySy, 110402), polyclonal chicken anti-MAP2 (1:10,000, Abcam, ab5392) and polyclonal rabbit anti-Munc18-1 (1:500, SySy, 116003) antibodies. After washing, cells were incubated for 1 h at room temperature with second antibodies conjugated to Alexa dyes (1:1,000, Molecular Probes, A20180, A20186) and washed again. Coverslips were mounted with Mowiol-Dabco and imaged with a confocal LSM510 microscope (Carl Zeiss) using a ×40 oil immersion objective with ×0.7 zoom at 1,024 × 1,024 pixels. Neuronal morphology and protein levels were analysed using a published automated image analysis routine[59].

**Data availability.** The data that support the findings of this study are available from the corresponding author upon reasonable request.

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

## Acknowledgements

We thank Robbert Zalm for expert help with cloning and virus production, Desiree Schut for preparing glia feeders and culturing neurons, Frank den Oudsten, Erik Ceelen, Joost Hoetjes, Joke Wortel, Christiaan van der Meer for breeding and genotyping mutant mice. We thank Dr S. van der Sluis for suggestions regarding statistical testing, Prof Nils Brose (Göttingen) for sharing *munc13-1/2* and *CAPS1/2* null mice, Prof Phyllis Hanson for valuable advice on NSF-interfering strategies and Prof Thomas Kuner (Heidelberg) for advice on NEM-interfering peptides. This work is supported by the European Union (ERC Advanced grant 322966; HEALTH-F2-2009-241498 EUROSPIN, and HEALTH-F2-2009-242167 SynSys to M.V.).

## Author contributions

E.H., L.N.C. and M.V. designed the research. E.H. performed most electrophysiology and all histology. K.W. and L.N.C. performed the initial electrophysiology on munc18-1/2SWAP neurons. K.W. performed ultrastructural morphometry. R.v.W. performed all the slice recordings. R.F.T. supervised/optimized cell biology and neuronal cultures. E.H., K.W., J.H.B. and L.N.C. analysed the data. E.H. and M.V. wrote the paper with input of all authors.

**Additional information**

**Competing interests:** The authors declare no competing financial interests.

