## [Peer Review File · Nature Communications]

Reviewers' expertise:

Reviewer #1: presynaptic release machinery;

Reviewer #2: presynaptic release machinery.

Reviewers' comments:

Reviewer #1 (Remarks to the Author):

The manuscript by He et al., from the Verhage laboratory is a solid piece of work examining the mechanisms of synaptic transmission. The speed and precision of calcium-evoked neurotransmitter release remains a fundamental process that has yet to be fully understood at a mechanistic level. Furthermore, it is increasingly apparent that diseases of the nervous system center upon defects at synapses. As such, this work should have broad appeal to the neuroscience community and is suitable for publication in this journal.

The experiments are well-performed and the effects that are reported are quite large. I find the interpretation of the experimental data reasonable. I advocate for rapid publication of this work. There are many ways that this study could be extended, but I do not see any experiments that 'must' be performed to verify the basis for the authors conclusions. It is a sound study with difficult and rather creative experiments.

Reviewer #2 (Remarks to the Author):

He et al show that replacing Munc18-1 by a non-neuronal isoform (Munc18-2) strongly reduces the size of the releasable vesicle pool at rest. The pool can be transiently increased by stimulation, but subsequently 'deprimes' within a few seconds. They also show that this instability of the vesicle pool can be overcome by blocking NSF. These data are highly interesting for two reasons: First, they are important for understanding the priming process. In particular the effect of the block of NSF indicates, that partially zippered SNARE complexes are vulnerable towards dissociation by NSF or else, that the primed state is dynamic (on/off) and a precursor of the primed state is vulnerable. Second, the finding that a non-neuronal form of Munc18 renders priming Ca^{++} -dependent (assuming that activity-dependence is mediated by Ca^{++}) provokes interesting questions regarding Ca^{++} -dependent priming in non-neuronal secretory cells and in other types of synapses: Tonic synapses in crayfish neuromuscular junction were best described by the assumption of an unstable primed vesicle state (Pan and Zucker, 2009, Neuron 62: 539). Hippocampal mossy fiber synapses show a runup during repetitive stimulation (Delvendahl et al, 2013, j. Physiol. 591: 3179), which is generally interpreted as a slow form of facilitation, but may indeed be the recruitment of a labile vesicle pool very similar to what is shown here in the presence of Munc18-2. Last not least, the releasable pool of granules in adrenal chromaffin cells is well-known to be strongly Ca^{++} -dependent and even shows 'depriming' (Dinkelacker et al, 2000, J. Neurosc. 15: 8377). It would be interesting to see whether the Munc18 expression pattern is different in these cells relative to that of the hippocampal cultures. Unfortunately none of these findings are discussed.

I have minor points of criticism:

Abstract: Strictly speaking the second sentence of the Abstract may be misinterpreted in the sense that miraculously replacing Munc18-1 by Munc18-2 rapidly leads to a fallback of primed vesicles. One more sentence may be required to make it clear that the priming under Munc18-2 is activity-dependent.

Figures: The panels are referred with capital letters in the text up to line 117, while they are labeled with lower case letters in the figures.

Line 133: 'did no longer after' may be better replaced by 'no longer did so after'

Line 154: Supplementary Figure 3ab is referred to, but the data are found in SF 4

Fig 3 and 5: The yellow traces and letters are hardly visible in my printout.

Lines 156 to 163: this is unclear, the reference to Suppl. Fig 3c, d, e should probably be to Suppl. Fig 4g, h; again some panels are referred to in capital letters in the legend (E and F). All references to figures should be scrutinized for correctness.

Rebuttal manuscript NCOMMS-17-03172, He et al.

We thank the reviewers for their constructive and positive feedback on our manuscript, and for their appreciation of the scientific significance of our work. Below is a point-by-point response to each of the issues raised.

REVIEWER 1 puts forward no issues to rebuttal.

We appreciated the reviewer's effort of reviewing the paper. We are also grateful for his/her highly supportive comments.

REVIEWER 2 puts forward 2 comments and 4 textual/cosmetic adjustments

We appreciated the reviewer's effort of reviewing the paper. We are also grateful for his/her highly supportive comments and helpful suggestions.

Comments 1 The reviewer indicates a number implications of our work that we failed to discuss (previous suggestions on 'unstable primed state', unexplained run-up during repetitive stimulation, calcium dependent pool-size modulation in chromaffin cells). We are very grateful for these suggestions and have now expanded the discussion on this issue by expanding the 3d paragraph in the discussion (p11-12), citing the studies indicated by the reviewer and also several others. We have also addressed expression patterns of different Munc13 isoforms (although only a few studies are available).

Comments 2: The reviewer points out an unprecise statement of Munc18-2 priming function in the abstract. We agree with this. Indeed, priming in Munc18-1/2 SWAP neurons is calcium/activity dependent. We added "activity-dependent" in line 21.

Textual/cosmetic adjustments: The use of capital or lower case letters has been adjusted, the syntax error in line 133 is corrected, the reference of supplementary figure 4 is corrected, the yellow traces in Fig. 3 and 5 have been changed to light green and we have corrected the reference of supplementary Fig. 4.

Other adjustments: we have added an acknowledgement regarding help we received from Dr. Ph. Hanson, which we forgot to mention in the original version of the manuscript.

REVIEWERS' COMMENTS:

Reviewer #2 (Remarks to the Author):

The authors have successfully addressed my points of criticism. The paper is a very valuable piece of work